# Clinicians’ Perceptions of Parent-Child Arts Therapy with Children with Autism Spectrum Disorders: The Milman Center Experience

**DOI:** 10.3390/children9070980

**Published:** 2022-06-29

**Authors:** Michal Bitan, Dafna Regev

**Affiliations:** 1The School of Creative Arts Therapies, University of Haifa, Haifa 3498838, Israel; michalbitan10@gmail.com; 2The Emili Sagol Creative Arts Therapies Research Center, University of Haifa, Haifa 3498838, Israel

**Keywords:** parent-child arts therapy, children, Autism Spectrum Disorders, the creative arts

## Abstract

Different types of arts offer a wide variety of modes of nonverbal communication and expressive tools for children with Autism Spectrum Disorders (ASD). The present study was designed to characterize therapists’ perspectives on the implementation of a parent-child arts therapy model for children with ASD. Semi-structured interviews were conducted with 13 arts therapists who participated in the study. The thematic analysis (qualitative analysis) approach yielded seven themes: (1) Therapeutic goals. (2) Adjusting the therapeutic intervention. (3) The advantages of parent-child arts therapy. (4) Difficulties in parent-child arts therapy. (5) The unique contribution of the participants to parent-child arts therapy. (6) The different types of arts in the therapy room. (7) The arts therapists’ assessment of the progress of therapy. The discussion focuses on the four central components of parent-child arts therapy room: the child in therapy, the parent, the arts therapist, and the creative arts.

## 1. Introduction

Parent-child arts therapy associates the therapeutic models of parent-child psychotherapy and the creative arts therapies (which include visual art therapy, bibliotherapy, music therapy, drama therapy, psychodrama and dance and movement therapy. Each of these modalities implements its own specific art for therapeutic purposes) [1,2,3,4]. The present study describes creative arts therapies with children with Autism Spectrum Disorders (ASD) according to the parent-child psychotherapy model and is based on therapists’ perspectives and clinical experience.

### 1.1. Parent-Child Arts Therapy

Parent-child arts therapy is becoming more prevalent, and several qualitative studies have been conducted to conceptualize the use of this type of therapy. The findings suggest that in parent-child arts therapy, spontaneous art-making allows the parent and child to express feelings, unconscious fears, desires, fantasies, and memories, thus reinforcing the bond between the parent and the child. The use of art materials expands the parents’ understanding of their child and enables them to develop a more reflective awareness of life experiences [3,4,5,6].

For example, Hassid conducted a quantitative study that examined the efficiency of parent-child art therapy in a group format (composed of several dyads) as a form of nonverbal communication in 5–8-year-olds who were referred to therapy. Twenty-two children were randomly divided into two groups: 10 children and their mothers received group parent-child art therapy and 12 children received group art therapy without the presence of the mothers. The findings pointed to a trend in the children in the experimental group toward improvement in social behavior and a significant improvement in certain measures of self-perception. There was also significant improvement in the mothers’ perceptions of their relationships with their children in the experimental group, as compared to the control group that only consisted of children involved in group art therapy [7]. In a preliminary study of parent-child dance and movement therapy, Weston [8] showed that the 10 participating mothers and children showed an improvement in their relationships and in terms of their communication. The mothers improved their parenting skills, and the children improved in terms of their behavioral measures. A recent study included a qualitative examination of the subjective perceptions of mothers in group mother-child dance and movement therapy [9]. In the quantitative part of the study the participants were randomly divided into two groups: 40 children in group mother-child dance and movement therapy and 40 children in group dance and movement therapy without their mothers. The findings showed that the mother-child group made significant improvement on some of the measures in comparison to the group of children without their mothers [10]. In a group parent-child music and dance and movement therapy [11], the participants (20 dyads) noted that they felt considerable improvement in their mutual understanding in terms of their general relationship and as a result of the intervention, and that they felt less tense and calmer. Moore [12] described parent-child drama therapy with parents and their adopted children (21 dyads) and noted that at the end of the intervention, the parents reported that they experienced greater mutual understanding as well as improved communication.

### 1.2. Parent-Child Arts Therapy with Children with ASD

Psychotherapists who use the parent-child model when working with children with ASD have described improvement in communication and in the relationship between children and parents [13]. They also observed improvement in the children’s ability to engage in symbolic play [14,15], the way the children felt that they had been understood, and in terms of support for the parents [16]. A recent study examined the comparative effectiveness of parent-child psychotherapy (Parent–Child Interaction Therapy) for teens with ASD (14 dyads) and without ASD (14 dyads). The results indicated that parent-child psychotherapy significantly improved parent-reported disruptive behavior in children with ASD at levels comparable to children without ASD. Improvements in ASD-related symptoms were also noted for teens with ASD [17].

The different types of arts offer a wide variety of modes of nonverbal communication and expressive tools for children with ASD. Clinical papers indicate that arts therapy enables children with ASD to engage in a nonverbal form of self-expression that can convey their experiences in a non-threatening way because it does not require the use of words, which is often complex for children with ASD [18]. Art-based interventions activate the senses of sight, hearing, touch and smell. In addition, art therapists use different materials with different colors and textures. These can be used as a means of sensory regulation for the child. The activities can be adapted to individual clients in a way that gives them a sense of security [19,20,21,22,23]. In the field of music therapy, preliminary quantitative studies have examined the significance of making music together for children with ASD. For example, a study examining the dyadic work of 10 children with ASD with their music therapist used exploratory factor analysis to show that dyadic drum playing was related to social skills [24]. Another quantitative study in music therapy examined the shared interactions between parents and their child with disabilities or developmental delays. Twenty-six parent-child dyads participated in this pretest-posttest within-subject single-group study. The parent-child dyads participated in a home-based music therapy program. The results showed that the parents’ positive physical and verbal responses, as well as the children’s positive verbal initiations, increased significantly pre- to post-intervention; however, the children’s positive physical initiations did not increase significantly. Parent-child synchrony also improved significantly pre- to post-intervention [25].

There is scant literature on parent-child arts therapy and most publications are based on case studies or clinical presentations that suggest approaches to working with this population. This underscores the need for research in this new field.

Parent-child arts therapy with children with ASD at the Milman Center in Israel (“One of the unique features of the Milman Center is its intensive parent-child therapy program with a psycho-developmental approach, placing the parents in the center of the therapeutic process, which in turn aims to widen the range of the child’s development. The therapists co-work with the parents to develop the child’s motor, sensory and language skills, as well as social communication and emotional development skills. The parents are an essential element of the therapy process and receive continual supervision and emotional support from the Milman team throughout the program.” Retrieved 19.11.21 from: https://www.milman-center.org.il/the-milman-center) treats children with ADS from the time of diagnosis (at approximately 18 months) until the children start school. The children go to the center twice a week for parent-child sessions with a multidisciplinary team. During each session, the children are accompanied by either their mother or their father who are seen by different professional teams. Specifically, each child receives two parent-child arts therapy sessions per week—one with father and one with mother—usually with different arts therapists. In these parent-child arts-therapy sessions, parent and child are invited to engage in various types of arts depending on the therapist’s specialization. They also engage in different kinds of arts, play or verbal exchanges according to the needs of each dyad. The present study was designed to characterize the parent-child arts therapy model at the Milman Center from the perspective of the therapists and their clinical experience. It examined the therapeutic goals, the working methods and interventions, the impact and value of using different types of arts, the difficulties and challenges experienced by the clients, and the therapists’ assessment of the effectiveness of this type of therapy as well as the needs of parents and children in therapy. These topics are explored below to contribute to developing an effective working model that can lead to the design of an applicable intervention method for children with ASD elsewhere.

## 2. Materials and Methods

### 2.1. Study Design

The study design was qualitative. This method draws on a phenomenological philosophical conception that aims to better understand the participants’ inner experiences without making prior assumptions. Qualitative research asks interviewees to describe their experiences in an authentic way and is oriented towards maximum expression to the interviewees’ views to gain a sense of their life experiences, in the context of the researched experience [26]. Specifically, this qualitative study was conducted according to the principles of the thematic analysis [27]. Thematic analysis is a method for identifying, analyzing and interpreting patterns of meanings or ‘themes’ in qualitative data. It is characterized by theoretical flexibility and organic processes of coding and theme development.

### 2.2. Participants

Thirteen arts therapists (M.A.) participated in the study (12 women and 1 man) including nine art therapists, three dance and movement therapists and one music therapist. An invitation to participate in the study was sent to all 28 arts therapists who were and are still working at the Milman Center in Haifa and Carmiel. All are trained to work according to the parent-child arts therapy model with children with ASD at the Center. When the interviews were conducted, the therapists ranged in age from 28 to 50 (M = 38.92; SD = 6.57). The therapists had work experience ranging from one year to thirteen years (M = 5.54; SD = 4.07). Specifically, at the Milman Center, this ranged from one year to thirteen years (M = 4.62; SD = 3.57). The children in therapy ranged in age from two and a half to seven years. Each arts therapist treats between three and twelve clients a year.

### 2.3. Data Collection

The researchers contacted the Milman Center to inquire whether the management was interested in taking part in this study. The Director of all the creatives arts therapists then informed all the arts therapists of the request, explained the goals of the study, and asked whether they would be willing to be interviewed. All 28 arts therapists working at the Milman Center were contacted. Thirteen agreed to take part in the study. The researchers decided to interview all the arts therapists who were interested to enable as wide a range of voices as possible. The therapists who were interested in participating responded to the Director who transmitted their names and contact details to the first author. The first author then contacted these arts therapists and again explained the goals of the study. The interviews were conducted between June 2020 and December 2020 individually in-person, at locations requested by the therapists.

The interviews were based on an interview guide (See Appendix A) that allowed the therapists to express their subjective opinions based on their practical experience [28]. The arts therapists talked about their working methods, the involvement of each participant in therapy, the success or lack of success they had experienced and the reasons in their opinion for them. They expressed their viewpoints on parent-child arts therapy including the goals they wanted to achieve, the importance for the client of including arts in the intervention, the difficulties and challenges they faced, the advantages of incorporating the arts, the advantages and disadvantages of including parents in the process, and finally their perceptions of the effectiveness of this therapeutic model.

### 2.4. Data Analysis

The data processing and analysis of the interviews was carried out according to the six stages of thematic analysis [27]. In the first stage, the interviews were transcribed and reviewed. In the second stage, primary themes and concepts that repeatedly appeared in the data were encoded. In the third stage, central themes were generated from the codes that emerged in the data, and in the fourth stage, links were defined according to the themes. In the fifth stage, central topics were defined and labeled that allowed for a generalization of each topic. The sixth stage consisted of writing a report that succinctly described the insights drawn from the data. Table 1 illustrates the work process on the first theme.

In the descriptions below, the phrase ‘absolute majority’ refers to 10–13 interviewees, the term ‘most’ refers to 8–9 interviewees and the phrase ‘about half ‘ refers to 6–7 interviewees. When a number of interviewees is mentioned, this is done to emphasize a certain point [29].

### 2.5. Rigor

At the end of the data processing and analysis process, the findings were presented to the professional team at the Milman Center. At the end of the presentation, the team was asked to comment and provide their opinions, a method proposed by Hill and colleagues [29]. Their opinions were integrated into the final section of the findings.

### 2.6. Ethical Considerations

The interviews began after a review of the ethical considerations of the study and the signing of an informed consent form. In addition, the researcher asked for permission to mention the arts therapists in future publications. Eleven therapists agreed. The interviews lasted between 56 and 95 minutes. The interviews were recorded and transcribed for the purpose of processing and analyzing the data. They were then sent to each therapist for approval. This study was approved by the ethics committee of the Faculty of Welfare and Health at the University of Haifa (107/20).

## 3. Results

### 3.1. Therapeutic Goals

#### 3.1.1. Mapping the Need and Setting the Goals

Most arts therapists stated that the goals of therapy are defined after mapping the personal needs of the individual clients according to their abilities and level of functioning: “We prepare a sort of diagnostic map of capabilities and difficulties.” Two arts therapists emphasized the children’s connection to their parents: “How the child and parent communicate, what their relationship is like, how the parent experiences and understands the child, how the parent understands what the child means… To understand their relationship.” They also noted that this was achieved through dialogue with the parents: “They have the space to talk, and they say what they feel their child needs.”

#### 3.1.2. The Primary Goals

##### Working on the Relationship

The absolute majority of arts therapists stated that the main goal of therapy is working on relationships: “This is obvious: creating a relationship, in ASD it’s always the relationship.” Two therapists mentioned the goal of opening the relationship to a triad: “Working on the ability to play in a trio, not just as a pair.”

##### Expanding and Developing the Element of Play

About half of the arts therapists mentioned the expansion of play: “I want to expand the element of play, because play is very, very, limited,” and developing mutual play: “Usually the main goal is to achieve mutual play, the ability to contain the other, to realize the presence of the other as a subject, not as an object.”

##### Working on Separateness between Parent and Child

Four arts therapists noted that alongside working on the relationship, there is also the work on separateness: “We noticed that the relationship became very symbiotic. As part of the process, we tried to enable the mother to work on her own surface next to her daughter.” Four arts therapists related to the importance of initiative and self-expression as a means of self-definition: “I go through a process of choosing colors with them…”

##### Working on Sensory Regulation

Four arts therapists related to the difficulties these children have in terms of regulation and the need to relate to these difficulties during therapy: “I tell them … You can apply some paint but try not to squeeze out too much. Then they get to a stage where they know to squeeze out a little bit of paint, and not squeeze out all the paint.”

##### Emotional Expression

Three arts therapists related to the possibility of developing emotional expression during the therapeutic session: “There was one client we wanted to expose to a large range of emotions. We wanted him to understand and internalize and maybe even share with us what he was feeling, so we prepared a kind of circle of emotions.”

### 3.2. Adjusting the Therapeutic Intervention

#### 3.2.1. Adjusting the Room and Setting

Most arts therapists stated that they prepare basic and accessible materials, games or musical instruments: “Usually there are a few materials that I prepare ahead of time: markers, sometimes watercolors. There are also games or musical instruments present.” About half the arts therapists noted that they devote the first sessions to observing the children and learning what interests them: “In the first sessions my goal, and I share this with the parents, is to observe the child, watch where s/he goes, what s/he looks for, what s/he likes and what elicits his/her curiosity.” The absolute majority of arts therapists remarked that after they became familiar with the dyad, they offered materials or interventions according to the goals of therapy: “I think that later it depends entirely on the therapeutic theme that I implement with this child.” Four arts therapists mentioned that they use augmentative and alternative communication symbols to illustrate an activity, the materials or the emotions: “Near the table there is a board with basic pictures, first of emotions… There is a folder with pictures of all the activities in the room…”

#### 3.2.2. Structure of the Session

Most arts therapists reported that the beginning and the end of the sessions are structured: “This framework is important, it has a relatively standard start, and a kind of closure”, but the main part of the session is more open, and the child is free to choose the activities and the materials: “In my room there is a closet that is left ajar so you can see what’s inside… Then they can actually take things out or ask for them.” Five arts therapists indicated explicitly that they let the children orient the course of the session. The arts therapist and the parent are there with them, based on the premise that where they feel good is where they will feel motivated: “To see where the children go and try to be there with them, understanding that when these children are motivated there is a much better chance for communication.”

#### 3.2.3. Intensive Intervention of the Arts Therapist

All the arts therapists remarked that they were active participants in the parent-child arts therapy process for children with ASD: “In parent-child arts therapy with these children I am definitely a very active participant.” They do so mostly to encourage the parents to be active in the therapeutic sessions and especially when the parent is experiencing difficulties: “Often when the parent… lacks energy or sits in the corner or doesn’t know what to do, I take the role of the parent and demonstrate how it should work,” or in order to connect with the children’s experience: “I try to personally experience their movements. I often invite the parent to try too.”

#### 3.2.4. Understanding and Mirroring the Meaningful Content That Emerges in Parent-Child Arts Therapy

All the arts therapists remarked that they help in mirroring and conceptualizing the emotions of the members of the dyad: “I try to mirror things that everyone does or things that I believe they are thinking or feeling” even if this involves mirroring the child’s emotions for him/herself: “One day a child suddenly realized that his mother was not there, that she had left and he was astonished, and I tried to mirror to him what he was feeling, that he might be a little apprehensive that his mother had left, that she was not here in the room,” or mirroring the parent’s behavior to the child: “When the father doesn’t want to do something, I can say: ‘Well it seems that daddy is not so interested in playing this game, he is bored.’” About half of the arts therapists noted that one of their responsibilities is to understand the content expressed when the child plays: “In a memory game a lot of things surface where you can learn about the child, the parent and the relationships,” or from the child’s repetitive speech: “When I discovered it and the parents saw it with me… He is not just repeating what he had just heard like a formula; he really wants to say something about himself, there is a deeper and more interesting thought process there…”

#### 3.2.5. When Parent-Child Arts Therapy Is Not Appropriate

Five arts therapists addressed the issue of situations in which parent-child arts therapy may not be appropriate. They described cases where the relationship is very complex and does not allow for growth: “There are cases where the dyad… itself is so complex and has all sorts of problematic aspects, that it does not leave room for growth.” They mentioned cases where the parent did not attend sessions regularly or behaved in ways that were counterproductive for the child: “Sometimes there is unsuitable language, inappropriate attitudes, taking control of the therapy, sometimes actually sabotaging therapy.”

### 3.3. The Advantages of Parent-Child Arts Therapy

#### 3.3.1. The Benefits of Parent-Child Arts Therapy Specifically for Children with ASD and Their Parents

Most arts therapists related directly to the relationship between parent-child arts therapy and children with ASD: “I believe that for young children, it’s parent-child psychotherapy, so definitely for children with ASD.” They argued that for children with ASD, attending therapy with their parents can pave the way to including other people later on in life: “For lower functioning children… I feel that the parents are sometimes really the key. They are definitely the most significant figures in the child’s life and at this time these children cannot include another figure.”

#### 3.3.2. Understanding the Child through the Mutual Learning of the Parent and the Therapist

Most arts therapists indicated that they learn about the children by watching their interactions with the parent: “There was something much richer… in my ability to understand the child when I see him through his interaction with his parent.” They described how the parents help them by interpreting and explaining what the child is doing, which makes it easier for them to understand the child better: “When the parent is there, s/he helps you understand the child, explains things… and it makes it easier for me to get to know the child.”

#### 3.3.3. Strengthening the Connection between the Dyad Members and Broadening the Support Circle

All the arts therapists mentioned the importance of working together with the parents due to the dependence of the young child on the relationship with them: “Parental presence… because the parent is naturally, especially in the early years, such a central figure.” The arts therapists noted that since the parents participate fully in the therapeutic process, they feel they make a significant contribution: “I give them space to express their opinions and to feel significant in the thinking process.” The parents learn to accept the children with their difficulties: “It can help parents feel closer to their children, love them more, accept them more.” Most arts therapists remarked that in parent-child arts therapy, where the parent plays an active role in the process, the effect of therapy is broadened beyond the session itself: “They can take away many tips from what we did in the room and continue to use them in their daily lives at home, in kindergarten,” and the parent helps the child make the connection between therapy and daily life: “When parents become part of the therapy… the children suddenly grasp that there is a connection between the world when they are in kindergarten and the world when they are at home.” Arts therapy also helps the parent find solutions to difficult situations: “For my part, I encourage thinking—why is he doing this… to understand that maybe, if you understand why, you can help.”

### 3.4. Difficulties in Parent-Child Arts Therapy

#### 3.4.1. The Difficulty in Adapting the Model to This Population

Five arts therapists described the difficulties involved in applying parent-child arts therapy to children with ASD: “The issue of a parent of a child with ASD, the way you work with them, that is the focus. The parent-child arts therapy changes drastically.” Most of the difficulties described occurred in the first year of therapy in terms of level of knowledge: “I laugh because after being there for a whole year I am only now beginning to understand a little bit…”, and from the feeling of being overwhelmed: “You know, it was just going in that was so overwhelming, it was like a new world, both the ASD and the parent-child arts therapy.”

#### 3.4.2. The Feeling of Failure and Self-Criticism

All the arts therapists described their experiences of failure when they felt that therapy did not progress: “If you just play, I feel…well, I don’t know, well…you know… a little redundant. Wait a moment, what am I? A therapist or just there?” Most arts therapists used the word “boredom” and sometimes even harsher words to describe their difficulties in therapy: “It is really this huge emptiness and the boredom… and sort of death, of…there is nothing here.” These hard feelings were connected to repetitiveness, which raised many questions: “Children with ASD often engage in play that is repetitive and empty…what does it mean for the child? And what is therapeutic here?” Four arts therapists described specific dyads where they felt redundant and even a burden: “I felt I was incapable, that I don’t know how to be a therapist, that I am not interesting, I am not funny… I felt I was ineffectual in this dyad.” Five arts therapists indicated their difficulties especially during their early years, when the presence of another adult in the room caused their levels of self-criticism to soar: “I think that first of all there is another adult who watches you work in the room, and it takes you right away to a place… of kind of self-criticism. How did I do, how was my work, how does s/he see me?”

#### 3.4.3. Frustration and Difficulties in Instances of Lack of Cooperation and Connection between Members of the Dyad and the Therapist

Most arts therapists addressed their feelings of frustration when they fail to motivate the parent to take an active part in the therapy: “There are also families who are more challenging... Their engagement requires us to invest much more energy in therapy and it also often frustrates us.” Five arts therapists mentioned their difficulties and feelings that they had not been able to connect with the child: “Especially in communication disabilities... You feel the difficulty of therapy or the relationship... You have been treating the child for so long and you do not feel that there is a therapeutic alliance.”

#### 3.4.4. Anger and Frustration at Difficulties in Child-Parent Communication

Most arts therapists noted a feeling of frustration when difficulties arise in the quality of communication between the parent and the child: “The mother… read a story to her child and she… didn’t really read… my heart was torn a little… It seemed to me that the communication between them was flawed, and it was difficult for me.” In these cases, the therapists were torn between the parent and the child and felt anger: “A lot of times I feel torn between the two of them, I see the distress of the daughter, and I want the mother to see it too… Sometimes I feel that I get a little angry at the mother…” Three arts therapists remarked that in spite of their difficulties, they tried to understand the parents: “I have to always remember to see both the child and the parent and understand their very sensitive and difficult place, in order to be there with it.”

#### 3.4.5. Difficulties with Parents’ Criticism

About half of the arts therapists described their difficulties with criticism from the parents on a personal level or in relation to the arts therapy profession: “The principal asked me something about the therapy… she said: ‘Because the mother said that she doesn’t quite understand what goes on in your room,’ something like that.” Four arts therapists mentioned insulting comments they received from parents during therapy: “And then he comes out and tells me ‘What was that… Today was really boring’ … I felt like it was a kind of a slap in the face.” On the other hand, two therapists related to this issue and thought that it was possibly caused by arts therapists’ difficulties explaining the basis of their therapy: “Most of us find it really difficult to explain… To conceptualize our therapy. It works, but it is difficult for us to make it clear and put the parents’ minds at ease.”

#### 3.4.6. Difficulty in Accepting That There Is Not Much Use of the Arts in the Therapy

About half of the arts therapists described the disappointment they felt when they realized that the arts are not always used during the intervention in parent-child arts therapy: “It was a real crisis for me in the beginning, I thought that, look, I am an arts therapist and therapy should be with materials only… But you see that some children and sometimes even parents don’t relate to it…”

### 3.5. The Unique Contribution of The Participants to Parent-Child Arts Therapy

#### 3.5.1. The Children’s Role in Therapy

Most arts therapists remarked that the participation of these children depends on their level of functioning: “I worked with very low functioning children and it was very difficult to get a reaction from them. I also worked with high functioning children who would come into the room and immediately initiate play together.” Most of them noted that in order to feel secure, children will often turn first to a familiar activity: “The children go to a game they know, something they are used to doing.” Most arts therapists noted that in parent-child arts therapy, the children express their needs to the parent: “Children’s play is often their way to tell the parent what preoccupies them, what they would like to change, what they want from the parent.” The arts therapists stated that during parent-child arts therapy sessions the child and the parent share an experience, even though sometimes it can be less than pleasant: “Her communication with him was through very basic games… At first, he would indicate that it was a little too much for him… but little by little, this was their way to communicate.” This is how the child learns to adapt to the parent. The therapists remarked that the children benefit from the parents’ presence as they continue to live their lives with the parent outside the arts therapy room: “This is the most important thing because in the end, they go home with the child and their interactions with the child continue throughout the day.” Most arts therapists mentioned that the bond between parent and child becomes stronger and more secure through the process of learning, and through the child’s ability to express him/herself: “The bond between parent and child becomes much stronger and more secure.”

#### 3.5.2. The Parents’ Role in Therapy

Most arts therapists noted the parents’ difficulties coping with their child’s disability: “For a parent it is unbearable, the parent comes into the room and sees the autism and sees this repetitiveness, the rigidity and the fact that nothing changes.” Most arts therapists described how during the encounter with the parents, the need for professional mediation becomes clear: “Parenting a child with special needs, especially with ASD, does not come naturally, it requires a learning process, and it often needs this professional mediation.” Most arts therapists said they understood the parents’ difficulties in attending therapy at the Milman Center in terms of scheduling: “It’s once a week and she comes for four hours. It’s a lot”, and in terms of coping: “You need to go on taking care of the child and watch him/her with the other children, and cope with the behavior patterns that may be unanticipated and not easy”, and from an emotional standpoint: “It’s so easy to feel criticized, so easy to feel I am not doing the right thing as a parent who comes to parent-child arts therapy.” Most arts therapists were able to relate to the parents’ difficulties in taking part in therapy: “For some of the parents it is hard to be in a room with another adult who can see their relationship very closely; they feel exposed and it’s not simple.” Some of them prefer to sit on the side and watch or detach themselves by using their phone: “There is one mother who often says: ‘No, I will sit on the side, I will watch and learn as an observer’ when asked to play” Sometimes parents struggle to deal with the mess in the arts therapy room: “I gave him gouache and told her that… I prepared her of course, that it will be messy and to be ready for it,” and coping with the boredom of the repetitive play of the child: “The father sometimes tries to encourage the child to play different games.” Four therapists remarked that with time, most parents learn to become more playful: “Little by little, they understand what we do here and how you do it and they can become more playful.”

All the arts therapists said that in parent-child arts therapy, the parents discovered new aspects of the child’s character and the child’s new abilities: “The mother said: ‘Wow this is new, this is the first time he has shared his drawing with me.’” The parent learns to understand and accept the child: “Because the parent goes through a process and participates in so many therapy sessions and learns to accept the child more and to understand him/her better… everybody benefits,” and simply to love him/her: “It’s very significant for parents to be able to connect and to love their child the way they are… Because it is not self-evident, especially when children have difficulties communicating.”

#### 3.5.3. The Arts Therapists’ Role in Therapy

All the arts therapists related to their need to understand the dyad and how to approach it in a way that can provide an appropriate response: “One moment, I see something good is happening between them so I back off, or one moment I have something good going with the child so I tell the mother go in this direction, you try it.” About half the therapists talked about how important it is to find their most beneficial place in relation to the child: “I now try to find how I can be with him, so that I feel him and he feels me and we play together and relate to each other even when he turns his back to me… so it will have meaning,” but in a way that the parent will not feel threatened: “With some parents I feel the need… to keep my distance more, that they might feel threatened maybe… if I get too close… and there are parents who I feel allow me from the very beginning to be very actively involved.”

Most arts therapists mentioned that they focus on forming a better connection between members of the dyad. They use modeling to do so: “I actually tried to show her how I play with him, how I am with him in terms of what he wants.” Four arts therapists noted that at the beginning of the relationship, they concentrated on forming an initial connection with the children: “In the initial stages I… would focus more on the child than on the parent, to form an initial acquaintance…” whereas two other therapists thought that the first bond should be with the parent: “I think that the initial bond with the parent is even more important than connecting first with the child.” Most arts therapists remarked that their role is to support the parents: “The process that happened there… to help her understand that it is not because she is not good, but rather take a moment to look at the strengths.” Four arts therapists stressed that the parents must be respected and parental authority acknowledged while maintaining parental competency: “This issue of recognizing parental authority… they are the experts when it comes to their child and I try to understand what they know and understand…” Three arts therapists indicated that they encourage the parents to share their difficulties: “Talk about it, about the difficulty…Encourage the mother to express interest… To contain the difficulty, to validate what emerges in this dyad.”

### 3.6. The Different Types of Arts in the Therapy Room

#### 3.6.1. Methods of Integrating the Arts into Therapy

Most arts therapists mentioned that they would like to find ways to integrate the arts into the therapeutic sessions. Dance and movement therapists noted it was easy for them to integrate their modality because movement is always present: “The advantage is that it is really there all the time… It can happen the minute you start throwing a ball or lying on a mattress.” These therapists attempt to explore these children’s movements and learn about their emotional state: “To try to understand his movements, what his body is telling me about the emotional experience.” The art therapists described how they get acquainted with these children by exposing them to the art materials. In the beginning, they only use a few materials to avoid overwhelming the child: “Naturally, in the beginning, a little bit, just to see if the child is not overwhelmed with what s/he sees.” They suggested that contact with art materials is part of self-understanding: “A possibility to investigate the body interacting and touching various things. It is another opportunity for the children to get to know themselves.” In music therapy, special effort was needed to integrate music into therapy: “With music …I felt that they were less open to it, that I was the one who introduced it and it was less their initiative,” and the different ways to include it: “Sometimes I would play something during a session and then I can attach a soundtrack to their dynamics. If, say, they start to fight, then suddenly the music is noisier and angrier.” Here too movement is integrated into the session: “Sometimes I strike a gong and then everyone freezes into some sort of sculpture.”

Arts therapists in all modalities remarked that not all the children like the arts: “You can see that there are children who don’t relate to it.” Some recoil from contact with the art materials: “Some children recoil when they come into contact with gouache and with certain textures of art materials”. Three arts therapists described children who engage in artmaking, but only for brief intervals: “But it is… you know, it’s three minutes…that’s it”. One of them believes that it generates anxiety: “I felt that it creates anxiety. There is an expectation for an outcome.” The therapists described how engaging in play can sometimes create an opportunity to work with the arts: “He played with animals and dipped them in paint and made a footprint… After a long period of not using art materials… Afterwards he continued creating with gouache, finger paints and glitter.” Six arts therapists noted that they do not always work in their modality but rather a different modality that the child relates to more.

#### 3.6.2. The Strengths of the Arts

All arts therapists described arts as an alternative language: “It’s a way to discover the rich world around us, that sometimes children with ASD kind of avoid touching,” that enables the creation of a playful space: “It involves creating something that is very playful, with art materials.” It also encourages closeness between the child and the therapist: “I feel that I can be part of these movements and it will connect us.” The arts therapists specified that arts make it possible to work on the issue of control: “It’s possible to play with themes of control through music… ‘now you play loudly and now softly’,” by regulating and channeling violence: “We can channel the violence to other materials rather than to the mother.” The arts therapists noted that working with the arts helps the parent share and connect to the experience of the child: “I see great importance in inviting the parent to draw near the child… This way the parents can relate to their children’s feelings when they draw.”

### 3.7. The Arts Therapists’ Assessment of the Progress of Therapy

#### 3.7.1. Progress in the Child’s Expressive Ability

Three arts therapists noted specific improvement in language abilities as a result of the therapeutic process: “It is really a long process… Suddenly there were words in the room.” Five arts therapists described an improvement in the verbal expression of emotions: “…There are still outbursts sometimes but little by little they subsided, little by little he would say I need a moment in the corner to calm down. He was able to verbalize it.” The same was true for expressing a desire: “This is something new that I didn’t see before… This ability to express wanting to play with only one person.”

#### 3.7.2. Progress in Terms of the Parents’ Acceptance of and Adaptation to the Child

Five arts therapists highlighted the importance of the changes experienced by the parents: “The ability of the parent to change, the willingness to attend and to deal with all this complex content, I think that is the most significant part.” They also mentioned its influence on the child’s acceptance and progress: “A very big change when he could make space for the violence and participate in the game… the child experiences that the father accepts him and wants to be with him… The child felt that he had legitimacy and that he is much more accepted.”

#### 3.7.3. Progress in the Ability to Be in a Relationship

All the arts therapists related to the change and progress in the relationship among the members of the dyad as observed through the changes in the children’s behavior: “It’s a process of a full year… to show this initiative, to spread the fabric and create some space for himself… and then little by little he invited the father and, in the end, invited me too,” and in the parents: “A very significant component was that the father simply joined in the game and allowed himself to let go of something that was very suppressed or to experience something new for himself.” Most arts therapists related to changes in the structure of play that develops into broadening the communication circles until finally the desire to play with other children develops: “Now when I see him during recess, he says: ‘Would you like to play with me? Would you like to ride with me in the car? Let’s drive, I can push you in the car.’”

## 4. Discussion

The present study described parent-child arts therapy from the point of view of experienced arts therapists who work according to this model at the Milman Center. The discussion focuses on the four central components present in the parent-child arts therapy room: the child in therapy, the parent, the arts therapist, and the creative arts.

The arts therapists described the child with ASD as the focus of parent-child arts therapy. The goal of therapy was determined by the therapists by assessing these children’s levels of functioning and their ability to be in a relationship. The arts therapists described the primary goals that first and foremost included working on the relationship between the child and the parent, in addition to other facets aimed at improving the quality of life of the child, such as expanding play and its development, working on separateness between parent and child, sensory regulation and emotional expression. Contact and communication emerged as the core of this parent-child arts therapy model. In other studies, as well and with other populations [4], arts therapists who used this model reported that they focused on the relationship between the child and the parent as a source and a starting point for creating change. This approach is even more pertinent with children with ASD whose major difficulty is related to relationships and communication [30]. Similar to articles that have described clinical work in the field of arts therapy with children with ASD [22,31], here too the arts therapists described how the children learn to express themselves and learn more ways to relate to the outside world. The presence of the parent allows for a broader understanding of the child and working together expands the child’s circles of communication for dyadic work and occasionally for triadic therapeutic work.

Most arts therapists related to the parents’ difficulties in coping with the child’s disability and vulnerability. These difficulties can manifest in different ways when the parents feel guilty and criticized or overwhelmed with emotional issues including depression. Some parents find it hard to cooperate and some find it difficult to establish contact with the child. Note that many parents contact the Milman Center shortly after the primary diagnosis when they are still processing it [21,32]. Regev and Snir [3] indicated that when a child is in distress, parents may feel that their parental competency is being tested. On the one hand, the parents are worried about the child, and on the other, the parents may feel overwhelmed with feelings of guilt and failure. In addition, most interviewees noted the parents’ difficulty, as they saw it, in taking an active part in therapy and sometimes in understanding its meaning. These findings correspond to results from other studies where parents described their initial difficulties in dealing with the artistic medium, which is less familiar and accessible to them. This caused them to feel awkward when using the art materials and engage in self-criticism, which required a change in thinking patterns on the part of the parent [9,33]. Nevertheless, some arts therapists related to the parents’ needs for intensive professional guidance and reported a shift that occurred in terms of parental participation as therapy progressed. This change relates to different aspects of parenting, but also to opportunities to become more playful and involved. Studies with a longitudinal design on parent-child psychotherapy with mothers and their children with ASD clearly point to the development of positive mutual emotions between the mother and the child that lead to improvement in self-regulation [34] and reduced disruptive behavior, while enhancing communication skills over the course of several sessions. Researchers have noted that the progress persisted even after a longer period of time [35,36].

All the arts therapists related to their need to find the best position for them within the dyad. This can be determined by understanding the needs of the dyad. The arts therapists discussed the changes they needed to make as a function of the interactions in the arts therapy room and their efforts to find the best way to involve parents in therapy. Kaplan and colleagues [37] emphasized the importance of finding the most beneficial position for the therapist in relationship to the dyad at all points in time. This underscores the need for therapists to understand the parental experience and adapt their position and interventions to the members of the dyad at all times [6,33,38]. The arts therapists noted that their relationships with the parents help them gain a fuller understanding of the child in therapy. Together with the parents, they can try to understand the cause and meaning of a particular behavior. In addition, all the interviewees noted that their role also involves mirroring and conceptualizing the emotions that emerge during therapy. Most arts therapists noted that they engage in the process of modeling they think is the most suitable for each child. This, according to them, can help the child be more completely understood, contained, and secure. The findings showed that most arts therapists believe that providing support to the parents and recognizing their authority is part of their role, as well as including the parents in the process, and encouraging them to find solutions. Similarly, several authors in the field of parent-child arts therapy have stated that the therapist should suspend judgmental attitudes and make an effort to understand the difficulties the parents face [3,21] by serving as a “good grandmother” [39]. The arts therapists noted their own complex feelings of failure, boredom, emptiness, and anger that arise during therapy along with feelings of self-criticism, which are often engendered by the presence of another adult in the arts therapy room.

Most arts therapists stated that they would like to find a way to integrate the arts into the therapy session, but about half noted that some children do not always want to engage in arts modalities. Most arts therapists indicated that in order to feel secure, children will usually turn to a familiar activity, and many prefer to play rather than do artmaking. Some arts therapists mentioned that it is important to understand the sensory profile of each child and that some children recoil from touching or handling the art materials. A few arts therapists discussed these children’s difficulty regulating their emotions, and one therapist suggested that even the thought of artmaking might cause a child to feel anxious. However, a few arts therapists described the act of playing as a way for these children to become familiar with the world of arts. Malchiodi [40] and Martin [21] related to the wealth and abundance of art materials and instruments that impact all the senses, but that can also be an emotional overwhelming experience for children with ASD. Dunn [41], an occupational therapist, found that children with ASD have different patterns of sensory processing. They may absorb sensual stimuli at an extremely slow pace, perhaps not even notice the stimulus, or alternatively, they may react very intensely and be unregulated. In this situation, some children will develop a strategy of avoiding engaging with their senses to facilitate their self-regulation. This has led several authors [21,40,41] to note the importance of planning the therapeutic space and finding the right balance between the different stimuli so that the child can remain in the arts therapy room and benefit from the session.

### Limitations and Suggestions for Further Research

This study has several limitations. It was based on the perception of arts therapists who work at the Milman Center using the parent-child arts therapy approach. It was therefore written from their point of view without exploring the points of view of the parents or children with respect to the content that emerged in therapy or its effects on them. In addition, due to the limited number of arts therapists working at the Milman Center, all types of modalities were examined together as a whole, which presents a problem when attempting to characterize a specific type of arts therapy. Future research could develop this topic by investigating the parents’ experiences during the process or exploring each modality in more detail. Finally, a longitudinal study could examine the effectiveness of parent-child arts therapy with this population post-therapy.

## 5. Conclusions

The field of parent-child psychotherapy has made considerable progress in recent decades. Arts therapists have also begun to adopt this therapeutic approach, which combines work with children and their parents as illustrated at the Milman Center, where arts therapists work with children with ASD and their parents. The present study was designed to map this process. The findings suggest that according to the arts therapists, integration of the arts can contribute to the parent-child relationship but needs to take the sensory profile and individual characteristics of each client into account.

## Figures and Tables

**Table 1 children-09-00980-t001:** Analyzing the first theme according to the thematic analysis.

Theme	Sub-Theme	Sub-Sub-Theme	Examples of Quotes from the Interviews
Therapeutic goals	Mapping the need and setting the goals		“We prepare a sort of diagnostic map of capabilities and difficulties.” “How the child and parent communicate, what their relationship is like, how the parent experiences and understands the child, how the parent understands what the child means… To understand their relationship.” “They have the space to talk, and they say what they feel their child needs.”
	The primary goals	Working on the relationship	“This is obvious: creating a relationship, in ASD it’s always the relationship.” “Working on the ability to play in a trio, not just as a pair.”
		Expanding and developing the element of play	“I want to expand the element of play, because play is very, very, limited.”
		Working on separation between parent and child	“We noticed that the relationship became very symbiotic. As part of the process, we tried to enable the mother to work on her own surface next to her daughter.” “I go through a process of choosing colors with them…”
		Working on sensory regulation	“I tell them … You can apply some paint but try not to squeeze out too much. Then they get to a stage where they know to squeeze out a little bit of paint, and not squeeze out all the paint.”
		Emotional expression	“There was one client we wanted to expose to a large range of emotions. We wanted him to understand and internalize and maybe even share with us what he was feeling, so we prepared a kind of circle of emotions.”

## Data Availability

The data are not publicly available due to ethical restrictions.

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
