# Peer review of "Clinicians’ Perceptions of Parent-Child Arts Therapy with Children with Autism Spectrum Disorders: The Milman Center Experience"

_children, 2022, doi:10.3390/children9070980_

Round 1

Reviewer 1 Report

Design: it is recommended to describe the research design. It is descriptive, phenomenological... the thematic analysis is the procedure that has been carried out for the analysis of the results, but it is not the design of the method. Participants: it is recommended to add a table with the sociodemographic data of the study participants 3. Data Collection: This section would include the date of collection and dropouts. It is recommended to specify when data saturation is reached.

Author Response

We would like to thank both reviewers for their important suggestions.

Design: it is recommended to describe the research design. It is descriptive, phenomenological... the thematic analysis is the procedure that has been carried out for the analysis of the results, but it is not the design of the method. 

Done.

Participants: it is recommended to add a table with the sociodemographic data of the study participants.

Since the State of Israel is very small, a table listing the participants would immediately identify them to the management and would show who agreed to participate and who did not, which we promised not to do as part of our ethics agreement. This is why we only describe the participants in general terms.

Data Collection: This section would include the date of collection and dropouts. It is recommended to specify when data saturation is reached.

We added the dates and numbers of arts therapists we contacted for this study (all the arts therapists working in at Milman Center) and those who agreed to participate. In this study, the decision as to the number of interviewees was not related to theoretical saturation but to the reality in the field – simply getting the largest sample possible of people who were willing to participate.

Reviewer 2 Report

Interesting read and definitely a great addition to the field!

Author Response

Thank you very much!